# Safety of FOLFIRI + Durvalumab +/− Tremelimumab in Second Line of Patients with Advanced Gastric Cancer: A Safety Run-In from the Randomized Phase II Study DURIGAST PRODIGE 59

**DOI:** 10.3390/biomedicines10051211

**Published:** 2022-05-23

**Authors:** Camille Evrard, Thomas Aparicio, Emilie Soularue, Karine Le Malicot, Jérôme Desramé, Damien Botsen, Farid El Hajbi, Daniel Gonzalez, Come Lepage, Olivier Bouché, David Tougeron

**Affiliations:** 1Service d’Oncologie Médicale, Centre Hospitalo-Universitaire de Poitiers, 86000 Poitiers, France; camille.evrard@chu-poitiers.fr; 2Service d’Hépato Gastro-Entérologie et de Cancérologie Digestive, Hôpital Saint Louis, 75010 Paris, France; thomas.aparicio@aphp.fr; 3Service d’Oncologie Médicale, Institute Mutualiste Montsouris, 75014 Paris, France; emilie.soularue@imm.fr; 4Fédération Francophone de Cancérologie Digestive, EPICAD INSERM LNC-UMR 1231, Université de Bourgogne Franche-Comté, 21000 Dijon, France; karine.le-malicot@u-bourgogne.fr (K.L.M.); daniel.gonzalez@u-bourgogne.fr (D.G.); come.lepage@u-bourgogne.fr (C.L.); 5Institut de Cancérologie, Hôpital Privé Jean Mermoz, 69008 Lyon, France; jerome.desrame@orange.fr; 6Service d’Hépato Gastro-Entérologie et de Cancérologie Digestive, Hôpital Universitaire de Reims, 51100 Reims, France; dbotsen@chu-reims.fr (D.B.); obouche@chu-reims.fr (O.B.); 7Service d’Hépato Gastro-Entérologie et de Cancérologie Digestive, Centre Oscar Lambret, 59000 Lille, France; elhajbi@o-lambret.fr; 8Service d’Hépato Gastro-Entérologie, Hôpital Universitaire de Dijon, 21000 Dijon, France; 9Service d’Hépato Gastro-Entérologie, Hôpital Universitaire de Poitiers, 86000 Poitiers, France; 10Faculté de Médecine et de Pharmacie, Université de Poitiers, 86000 Poitiers, France

**Keywords:** gastric cancer, gastro-oesophageal junction adenocarcinoma, safety run-in, immune checkpoint inhibitors, irinotecan

## Abstract

Efficacy of immune checkpoint inhibitors (ICI) as monotherapy in 2nd line treatment for gastric or gastro-oesophageal junction (GEJ) adenocarcinoma is low, with no evaluation of efficacy and safety of ICI combined with chemotherapy. The DURIGAST PRODIGE 59 study is a randomised, multicentre, phase II study designed to assess the efficacy and safety of the combination of FOLFIRI + Durvalumab +/− Tremelimumab as 2nd line treatment of patients with advanced gastric/GEJ adenocarcinoma. Here, we report data from the safety run-in phase with FOLFIRI Durvalumab (arm A) or FOLFIRI Durvalumab and Tremelimumab (arm B). Among the 11 patients included, 63.6% experienced at least one grade 3–4 adverse events (AEs) related to the treatment, most frequently neutropenia (36.4%). There was only one immune-related AE (grade 2 hyperthyroidism). Ten serious AEs were described among six patients, but only two were related to the treatment, due to the chemotherapy. One seizure epilepsy related to a brain metastasis was observed, but was not related by the investigator to the treatment. However, the Independent Data Monitoring Committee recommended brain imaging at inclusion. This safety run-in phase demonstrates an expected safety profile of FOLFIRI with Durvalumab +/− Tremelimumab combination allowing the randomised phase II.

## 1. Introduction

Overall survival (OS) of patients with advanced gastric or gastro-oesophageal junction (GEJ) adenocarcinoma remains short, from 10 to 15% at 5 years. Addition of docetaxel to chemotherapy doublet (Platinum salt and 5-Fluorouracil, 5FU) for first-line treatment in Human Epidermal Growth Factor Receptor-2 (HER2) negative tumours has improved not only OS, but also toxicity [1,2,3]. Second-line treatments using a taxane (docetaxel or paclitaxel) alone or combined with ramucirumab or irinotecan alone or combined with 5FU (FOLFIRI) have improved OS (from 4.0 to 9.5 months) and progression-free survival (PFS) (from 2.5 to 5.3 months) as compared to BSC alone (about 3 months) [4,5,6,7,8,9].

Recently, immune checkpoint inhibitors (ICI) have shown significant efficacy in advanced gastric/GEJ adenocarcinomas, especially anti-Program Death 1 (anti-PD1) and anti-Program Death-ligand 1 (anti-PD-L1) monoclonal antibodies combined with platinum-based chemotherapy in first-line setting [10,11]. The CheckMate-649 phase III trial has shown higher OS and PFS with the nivolumab plus chemotherapy (XELOX or FOLFOX) combination in the subgroup of patients with a PD-L1 combined positive score (CPS) ≥ 5 versus chemotherapy alone [11]. By contrast, recent phase III trials comparing ICI alone versus chemotherapy have shown no survival increase [11,12,13]. The phase I/II CheckMate-032 study has compared nivolumab versus nivolumab plus ipilimumab and objective response rate (ORR) reached 24% [14]. Combination of durvalumab (anti-PD-L1) and tremelimumab (anti-Cytotoxic T lymphocyte antigen 4, anti-CTLA4) in a phase Ib/II study showed a 6-month PFS of 20% [15]. To conclude, ICI alone has shown low efficacy, but a combination of ICI and ICI plus chemotherapy combination has shown promising results.

Data concerning safety and efficacy of ICI plus chemotherapy in second-line setting of metastatic gastric/GEJ adenocarcinoma are lacking. Indeed, the randomised DURIGAST PRODIGE 59 phase II trial has been designed to assess the efficacy and safety of FOLFIRI with durvalumab or durvalumab plus tremelimumab as second-line treatment in patients with advanced gastric/GEJ adenocarcinoma. A safety run-in was planned to detect early and acute toxicity given that there are no safety data available on the combination of FOLFIRI regimen and ICI.

## 2. Patients and Methods

### 2.1. Study Design

The DURIGAST PRODIGE 59 study (NCT 03959293) was a randomised, open-label, multicentre, non-comparative, phase II study conducted in France [16]. Patients with advanced gastric or GEJ adenocarcinoma, pre-treated with a fluoropyrimidine plus platinum salt +/− taxane (F + P ± T), were randomized 1:1 between FOLFIRI plus durvalumab (arm A) or FOLFIRI plus durvalumab plus tremelimumab (arm B). Due to a lack of data concerning the combination of ICIs plus FOLFIRI, a safety run-in phase was performed before the randomised phase II.

Written informed consent was obtained from all patients before treatment. The DURIGAST PRODIGE 59 trial was submitted for formal approval to the French Health Authorities (ANSM) and an independent Ethics Committee.

### 2.2. Study Objectives

The primary endpoint of the randomized phase II was the percentage of patients alive and without progression at 4 months in the two arms based on the Response Evaluation Criteria In Solid Tumours (RECIST) 1.1 score evaluated by the investigator.

The main secondary endpoints were OS, safety profile and health-related quality of life. Adverse events (AEs) were described according to National Cancer Institute—Common Terminology Criteria for Adverse Events version 4.0 (NCI-CTCAE v4.0).

### 2.3. Study Population

Main inclusion criteria were patients with histologically proven advanced unresectable gastric/GEJ (Siewert II or III) adenocarcinoma, with progression or intolerance after first-line chemotherapy with F + P ± T, Eastern Cooperative Oncology Group (ECOG)—Performance Status (PS) 0 or 1 and adequate organ function. Main non-inclusion criteria were active or prior documented autoimmune or inflammatory disorders, use of immunosuppressive/steroid medication within 14 days before the first dose of study drugs and known dihydropyrimidine dehydrogenase (DPD) enzyme deficiencies [16] (see study protocol, Appendix A).

### 2.4. Treatment Scheme and Modalities

In arm A and B, Durvalumab was administered at a dose of 1500 mg in 1-h IV infusion every 4 weeks (Figure 1). In arm B, Tremelimumab was administered at a dose of 75 mg in 1-h IV infusion before Durvalumab, every 4 weeks. Tremelimumab was administered for only 4 cycles.

FOLFIRI regimen combined folinic acid 400 mg/m^2^ by 2-h IV infusion, 5FU bolus 400 mg/m^2^ by 10-min IV infusion, continuous 5FU 2400 mg/m^2^ by 46-h IV infusion and Irinotecan at 180 mg/m^2^ in arm A or 150 mg/m^2^ only for the safety run-in phase in arm B, by 2-h IV infusion every 2 weeks.

Treatment was repeated every 2 weeks until disease progression, unacceptable toxicity, death, withdrawal of consent or patient refusal. Dose adjustment was based on toxicity according to standard guidelines for FOLFIRI. Dose reduction was not allowed for durvalumab and tremelimumab.

Patients were evaluated every 2 weeks with standard clinical examination and laboratory assessment. Morphological assessment with thoracic-abdominal-pelvic CT-scan according to RECIST 1.1 criteria was performed every 8 weeks. Adverse events were collected every 2 weeks on day 1 of each treatment cycle and reported using NCI-CTCAE v4.0.

### 2.5. Safety Run-In Analysis

Given that the safety profile of the FOLFIRI + Durvalumab +/− Tremelimumab combination has not been evaluated so far, a safety run-in phase was requested by French authorities (ANSM, “Agence nationale de sécurité du médicament”). A safety run-in phase with 2 steps was performed to evaluate first combination of FOLFIRI + durvalumab, and then the combination of FOLFIRI + durvalumab + tremelimumab.

There were no specific selection criteria in this safety run-in, inclusion and non-inclusion criteria were the same as the randomized phase II [16] (see study protocol, Appendix A). There was no placebo group in this open-labelled study.

A total of 11 patients were required in the safety run-in phase, which was limited to five expert centres with huge experience in the use of ICIs. At each step, inclusions were stopped, and when all patients had received at least 2 cycles of treatment, the safety analysis was done on all the safety data available at that date.

The first step of the safety run-in phase planned to enrol 5 patients treated with FOLFIRI (Irinotecan at 180 mg/m^2^) and durvalumab (Figure 1). If there was no safety issue, the second step was performed. The second step of safety run-in phase planned to enrol 6 patients randomised to receive FOLFIRI (Irinotecan at 180 mg/m^2^) and durvalumab versus FOLFIRI (Irinotecan at 150 mg/m^2^), durvalumab and tremelimumab, with 3 patients randomised per arm.

Safety data were reviewed by an Independent Data Monitoring Committee (IDMC). There were no pre-defined criteria to stop the study.

### 2.6. Statistical Analysis

Descriptive statistics and individual data were done and are presented by arms and on the whole population. Quantitative variables were described with means, medians, standard deviations (SD) or interquartile ranges (IQR). Qualitative variables were described as frequencies and percentages.

All statistical analyses were carried out using SAS software 9.4 (SAS Institute, Cary, NC, USA).

## 3. Results

### 3.1. Patient and Tumour Characteristics

Eleven patients were included in the safety run-in phase of DURIGAST PRODIGE 59 trial between 17 July 2019, and 3 March 2020, 8 patients in arm A and 3 in arm B. The median age was 71 years and 36.4% of patients were female (Table 1).

Most tumours were GEJ location (81.8%) and intestinal histological subtype (70.0%). All tumours were microsatellite stable (MSS) and/or proficient MisMatch Repair (pMMR). Most patients had synchronous metastasis (81.8%) and the most frequent metastatic sites were the liver (45.5%) and lymph nodes (54.5%).

Regarding prior chemotherapy regimen, most patients received a first-line regimen treatment with doublet of chemotherapy (F + P) (72.7%).

### 3.2. Adverse Events in Overall Population and in Each Arm

At data cut-off 3 June 2021, median follow-up was 19.3 (95%CI: 14.7-not reached) months and 8 patients had stopped the treatment. Median duration of treatment was 7.6 months [IQR: 3.1–17.0] in arm A and 3.1 months [IQR: 2.1–11.1] in arm B.

Seven patients (63.6%) experienced at least one grade 3 or 4 AE related to the treatment, 5 in arm A (62.5%) and 2 in arm B (66.7%), respectively (Table 2). In arm A, the main AEs involved 3 patients with grade 3/4 neutropenia (37.5%) and 2 patients with grade 3/4 sequelae peripheral sensory neuropathy (25.0%). In arm B, one patient had grade 3/4 nausea (33.3%), one grade 3/4 neutropenia (33.3%) and one grade 3/4 fatigue (33.3%). AEs not related to the treatment are described in Appendix A.

Among the 11 treated patients, six patients (four in arm A and two in arm B) had at least one serious adverse event (SAE) with a total 10 SAEs. Among these 10 SAEs, two were associated with chemotherapy: anorexia grade 3 requiring a feeding jejunostomy and grade 3 deterioration of performance status requiring hospitalization. Among the eight SAEs unrelated to the treatment, there were two pneumopathies, two dysphagia, two pyelonephritis, one perigastric abscess and one epileptic seizure. Concerning this last SAE, the patient had brain metastases that were unknown before the epileptic seizure.

Only one immune-related AE (irAE) was observed in arm B (grade 2 hyperthyroidism), spontaneously resolving after the end of tremelimumab.

### 3.3. Modification of Treatment Related to Toxicity

At the data cut-off, eight patients had definitely discontinued the treatment (72.7%), seven due to disease progression (87.5%) and one patient due to an infectious pneumonia not related to the treatment (Table 3). One patient in arm A first stopped irinotecan only due to multiple grade 2 adverse events (anorexia, asthenia, diarrhoea and anaemia) and five weeks later discontinued the treatment due to disease progression.

Concerning the chemotherapy, irinotecan doses were reduced at least one time for two patients, one in arm A (dose reduction of 55.3%) and one in arm B (dose reduction of 33.3%), both due to toxicity. 5FU bolus doses were reduced at least one time for 2 patients in arm A (dose reduction of 50% and 48.8%) and 5FU continuous doses were reduced at least one time for three patients, one in arm A (dose reduction of 33.3%) and two in arm B (dose reduction of 33.7% and 27.6%), all due to toxicity.

Concerning ICIs, for tremelimumab there was no dose reduction. One patient had the four cycles of tremelimumab, and two patients stopped tremelimumab due to progression after two and three cycles, respectively. For durvalumab, there was no dose reduction, and eight patients had stopped durvalumab (72.7%), seven due to progression and one because of a pulmonary infection.

### 3.4. Modification of the Protocol According to Safety Run-In Phase

Based on these safety results, the IDMC decided to continue the study. Nevertheless, even though the epileptic seizure is related to a brain metastasis, the IDMC considered that it was not possible to rule out the possibility that the ICI promoted cerebral oedema around the metastasis, thereby causing the epileptic seizure. The IDMC has recommended, for patient safety, that brain imaging be performed at inclusion to identify brain metastases. These conclusions have been sent to ANSM. While ANSM then agreed to open the randomised phase II trial on 14 August 2020, brain imaging was required at baseline by CT-scan. Patients with previously unknown and untreated brain metastases were not included.

The randomized phase II started on 27 August 2020 and the recruitment was closed on 8 June 2021; final results are expected by the end of 2022.

## 4. Discussion

This safety run-in phase of DURIGAST PRODIGE 59 evaluated for the first time the combination of chemotherapy plus anti-PD-L1 and anti-CTLA-4 in gastric/GEJ cancers and demonstrated an expected and acceptable safety profile.

There is no statistical hypothesis for this safety run-in phase, which was requested by the French authorities. A total of 11 patients is low to draw definitive conclusion of the safety of the combination of FOLFIRI + durvalumab + tremelimumab. The patient’s number was defined with the French authorities. There are some safety data with the combination of FOLFOX + durvalumab + tremelimumab [17] Indeed, we thought that the combination of FOLFIRI + durvalumab + tremelimumab will be safe. We mostly want to demonstrated the absence of severe diarrhoea with this combination in a small patient’s number in order to start as soon possible the randomized phase II, which is not the case among the 11 patients (no grade 3–4 diarrhoea). The whole randomized phase II will provide more robust safety results on a larger patient’s number since only 11 patients have been included in this safety run-in phase of DURIGAST PRODIGE 59.

Patient population in this safety run-in phase is the usual one for trials in advanced gastric/GEJ cancers with mainly GEJ location, mostly synchronous liver and lymph node metastasis. Grade 3/4 AEs related to the treatment were observed in 63.6% of patients, including neutropenia (36.4%), peripheral sensory neuropathy (18.2%), fatigue (18.2%), proteinuria (9.1%), nausea (9.1%) and anaemia (9.1%). Ten SAEs were reported, but only two SAEs were related to the treatment and due to the chemotherapy. No grade 3/4 AE or SAE was related to ICI. Peripheral sensory neuropathy reported in two patients is not a common toxicity of FOLFIRI regimen, but a sequel to first-line chemotherapy with platinum salt and, as is well-known, it can continue to worsen despite the stop of platinum salt during the second-line regimen. Other toxicities are those commonly observed with the FOLFIRI regimen. In the literature, for irinotecan-based regimen, common grade 3/4 AEs were neutropenia (28–39%), anaemia (7–30%), diarrhoea (14–26%) and vomiting (1–6%) [5,6,7]. One of the fears in the DURIGAST PRODIGE 59 study was digestive toxicity, especially diarrhoea, which could be associated with both toxicity of irinotecan and colitis due to ICI. Nevertheless, we did not observe any grade 3/4 diarrhoea.

DURIGAST PRODIGE 59 study is the first combination of chemotherapy and two ICIs. We already know that a combination of anti-PD-L1/anti-PD-1 and anti-CTLA-4 increase the rate of grade 3/4 immune-related AEs as compared to an anti-PD-L1/anti-PD-1 alone, from 10 to 25% [11,14,15,18]. In this safety run-in phase, we observed only one immune-related AE (grade 2 hyperthyroidism). With a combination of chemotherapy and ICI, grade 3/4 AEs were observed for 59% to 73% patients in recent trials in advanced gastric/GEJ cancers [10,11,12]. Most common grade 3/4 AEs were neutropenia (15–25%), anaemia (6–12%) and diarrhoea (6–8%). This rate is in accordance with our results with 63.6% of grade 3/4 AEs.

We observed three dose reductions of chemotherapy (5FU and/or Irinotecan) due to toxicity. All but one treatment stops were due to disease progression. These rates are in accordance with FOLFIRI regimen toxicities in gastric/GEJ adenocarcinomas [5,6,7].

As all AEs were expected with an acceptable safety profile of FOLFIRI plus durvalumab alone or combined with tremelimumab, the French regulatory authority agreed to start phase II of the DURIGAST PRODIGE 59 trial. Nevertheless, a brain CT-scan was required at baseline since an epileptic seizure related to a brain metastasis was observed. Recent studies suggest that ICIs have significant efficacy with an acceptable safety profile in treatment of brain metastases from various cancers, including non-small-cell lung cancer and melanoma [19,20,21]. A recent review concluded that there is no additional neurotoxicity in patients with primary or secondary brain tumours and so far there have been no specific concerns regarding the neurological tolerability of ICI in patients with brain tumours [22].

## 5. Conclusions

DURIGAST PRODIGE 59 study is the first study evaluating FOLFIRI plus ICIs as second-line treatment of advanced gastric/GEJ adenocarcinoma, which remains an unmet need. Even if the combination of nivolumab plus XELOX/FOLFOX is now the standard of care in the subgroup of patients with a CPS ≥ 5 in first-line treatment, it remains a major issue to evaluate chemotherapy plus ICIs in second-line setting [23]. In the DURIGAST PRODIGE 59 study, while no patient had previously received an ICI, the results will help to determine whether FOLFIRI plus anti-PD-L1 and CTLA-4 provides better survival as compared to FOLFIRI plus anti-PD-L1. Primary endpoint is PFS with the hypothesis of 70% of patients alive and without progression at 4 months in the FOLFIRI plus durvalumab and tremelimumab arm (H0:50%). With a risk α of 5%, a power of 85% and according to the binomial exact method, 94 patients will be included for the randomised phase II. It is mandatory to evaluate immunological parameters to define the patients’ responsiveness to ICIs. Indeed, planned ancillary studies, in a centralized review, will identify predictive biomarkers of efficacy including PD-L1 expression and others immune markers, microsatellite instability, immune scores, tumour mutation burden and microbiota. These results will help to define the best combination to evaluate in a phase III trial in a second-line setting and also to determine whether this combination should be evaluated in all-comers or sub-groups of patients with relevant biomarkers.

## Figures and Tables

**Figure 1 biomedicines-10-01211-f001:**
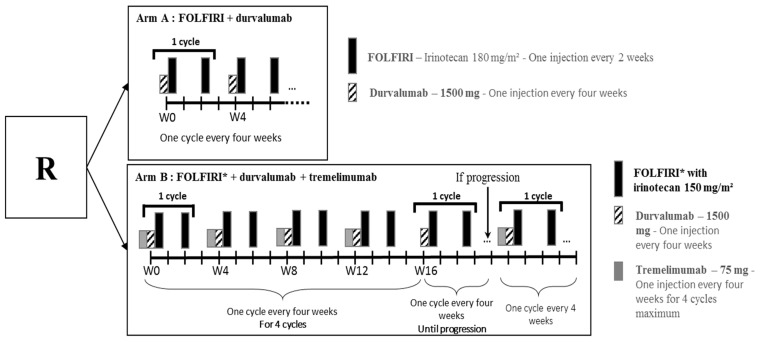
Treatment scheme of safety run-in phase.

**Table 1 biomedicines-10-01211-t001:** Patient and tumour characteristics.

Variables	All Patients (n = 11)	Arm A (Folfiri + Durvalumab) (n = 8)	Arm B (Folfiri + Durvalumab + Tremelimumab) (n = 3)
**Age (years, range)**	71 [42–78]	72 [55–78]	62 [42–70]
**Female (n, %)**	4 (36.4%)	3 (37.5%)	1 (33.3%)
**ECOG performance status (n, %)**			
0	4 (36.4%)	2 (25.0%)	2 (66.7%)
1	7 (63.6%)	6 (75.0%)	1 (33.3%)
**Body Mass Index (kg/m^2^, range)**	26 [21–28]	27 [21–28]	23 [23–26]
**Primary tumour site (n, %)**			
Gastro-oesophageal junction	9 (81.8%)	7 (87.5%)	2 (67.7%)
Stomach	2 (18.2%)	1 (12.5%)	1 (33.3%)
**Tumour subtype (Lauren classification) (n, %)**			
Intestinal type			
Diffuse type	7 (70.0%)	4 (57.1%)	3 (100%)
Unknown	3 (30.0%)	3 (42.9%)	0
	1	1	0
**Microsatellite instability**			
Deficient	0	0	0
Proficient	11 (100%)	8 (100%)	3 (100%)
**Delay of metastatic disease (n, %)**			
Metachronous	2 (18.2%)	1 (12.5%)	1 (33.3%)
Synchronous	9 (81.8%)	7 (87.5%)	2 (66.7%)
**Resection of primary tumour (n, %)**			
No	8 (72.7%)	6 (75.0%)	2 (66.7%)
Yes	3 (27.3%)	2 (25.0%)	1 (33.3%)
**Site of metastases (n, %)**		3 (37.5%)	2 (66.7%)
Liver	5 (45.5%)	1 (12.5%)	1 (33.3%)
Lung	2 (18.2%)	3 (37.5%)	0
Peritoneal carcinomatosis	3 (27.3%)	5 (62.5%)	1 (33.3%)
Lymph nodes	6 (54.5%)		
**Prior first-line chemotherapy regimen (n, %)**		6 (75.0%)	2 (66.7%)
Doublet regimen *		2 (25.0%)	1 (33.3%)
Triplet regimen **	8 (72.7%)		
	3 (27.3%)		

ECOG: Eastern Cooperative Oncology Group; * fluoropyrimidine + platinum salt; ** fluoropyrimidine + platinum salt + taxane.

**Table 2 biomedicines-10-01211-t002:** Patient and tumour characteristics.

n, %	Arm A (Folfiri + Durvalumab) (n = 8)		Arm B (Folfiri + Durvalumab + Tremelimumab) (n = 3)	
	Grade 1–2	Grade 3–4-5	Grade 1–2	Grade 3–4-5
**Patients with at least** **one adverse event**	**8 (100.0%)**	**5 (62.5%)**	**3 (100.0%)**	**2 (66.7%)**
**Skin and subcutaneous tissue disorders**	**5 (62.5%)**	-	**1 (33.3%)**	-
Pruritus	1 (12.5%)	-	-	-
Acneiform rash	2 (25.0%)	-	-	-
Dry skin	1 (12.5%)	-	1 (33.3%)	-
Palmar-plantar erythrodysesthesia	2 (25.0%)	-	-	-
**Renal and urinary disorders**	**-**	**1 (12.5%)**	**-**	**-**
Proteinuria	**-**	1 (12.5%)	**-**	**-**
**Nervous system disorders**	**4 (50.0%)**	**2 (25.0%)**	**1 (33.3%)**	**-**
Peripheral sensory neuropathy	4 (50.0%)	2 (25.0%)	1 (33.3%)	**-**
**Endocrine disorders**	**-**	**-**	**1 (33.3%)**	**-**
Hyperthyroidism	**-**	**-**	1 (33.3%)	**-**
**Gastrointestinal disorders**	**8 (100.0%)**	**-**	**2 (66.7%)**	**1 (33.3%)**
Constipation	1 (12.5%)	**-**	1 (33.3%)	**-**
Diarrhoea	6 (75.0%)	**-**	**-**	**-**
Dysgeusia			1 (33.3%)	**-**
Dyspepsia	3 (37.5%)	**-**	**-**	**-**
Mucositis	4 (50.0%)	**-**	**-**	**-**
Nausea	7 (87.5%)	**-**	**-**	1 (33.3%)
Vomiting	3 (37.5%)	**-**	1 (33.3%)	
**Blood and lymphatic system disorders**	**14 (100.0%)**	**4 (50.0%)**	**3 (100.0%)**	**1 (33.3%)**
Anaemia	5 (62.5%)	1 (12.5%)	3 (100%)	
Neutropenia	4 (50.0%)	3 (37.5%)	1 (33.3%)	1 (33.3%)
Thrombocytopenia	5 (62.5%)	**-**	**-**	**-**
**Musculoskeletal conditions**	**1 (12.5%)**	**-**	**2 (66.7%)**	**-**
Back pain	1 (12.5%)	**-**	2 (66.7%)	**-**
**General disorder**	**7 (87.5%)**	**1 (12.5%)**	**1 (33.3%)**	**1 (33.3%)**
Fatigue	5 (62.5%)	1 (12.5%)	1 (33.3%)	1 (33.3%)
Fever	1 (12.5%)	-	-	-
Anorexia	1 (12.5%)	-	-	-

The total of adverse events could be superior to the total number of patients since some patients could have more than one adverse event.

**Table 3 biomedicines-10-01211-t003:** Dose reduction and treatment stop.

n, %	n = 11	Definitive Discontinuation of Treatments (n = 8)			
	Dose Reduction for Toxicities	Treatment Stop due to Toxicities	Treatment Stop due to Progression	Treatment Stop Planned by the Protocol	Treatment Stop for Other Reason(s) *
**Irinotecan (n = 11)**	2 (18.2%)	1 (12.5%)	6 (75.0%)	0	1 (12.5%)
**5FU bolus (n = 11)**	2 (18.2%)	0	7 (87.5%)	0	1 (12.5%)
**Continuous 5FU (n = 11)**	3 (27.3%)	0	7 (87.5%)	0	1 (12.5%)
**Durvalumab (n = 11)**	0	0	7 (87.5%)	0	1 (12.5%)
**Tremelimumab (n = 3)**	0	0	2 (66.7%)	1 (33.3%)	0

* Infectious pneumonia not related to the treatment.

## Data Availability

Data can be made available upon reasonable request.

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
