# Peer review of "Safety of FOLFIRI + Durvalumab +/− Tremelimumab in Second Line of Patients with Advanced Gastric Cancer: A Safety Run-In from the Randomized Phase II Study DURIGAST PRODIGE 59"

_biomedicines, 2022, doi:10.3390/biomedicines10051211_

Round 1

Reviewer 1 Report

Biomedicines

The manuscript entitled “Safety of FOLFIRI + Durvalumab +/- Tremelimumab in second line of patients with advanced gastric cancer: a safety run-in from the randomized phase II study DURIGAST PRODIGE 59” by Camille Evrard et al, describes data from the safety run-in phase with FOLFIRI Durvalumab (arm A) or FOLFIRI Durvalumab and Tremelimumab (arm B). Among the 11 patients included, 63.6% experienced at least one grade 3-4 adverse events (AEs) related to the treatment, most 32 frequently neutropenia (36.4%). The authors concluded that this safety run-in phase demonstrates an expected safety profile of FOLFIRI with Durvalumab +/- Tremelimumab combination allowing the randomised phase II.

Overall, this is an interesting concept, however there are some important questions that needs to be addressed to support this study.

General comments:

  1. In the present study the DURIGAST PRODIGE 59 study (NCT 03959293) was a randomised, open-label, multicenter, non-comparative, phase II study conducted in France. Patients with advanced gastric or GEJ adenocarcinoma, pre-treated with a fluoropyrimidine plus platinum salt +/- taxane (F+P±T), were randomized 1:1 between FOLFIRI plus Durvalumab (arm A) or FOLFIRI plus Durvalumab plus Tremelimumab (arm B). Due to a lack of data concerning the combination of ICIs plus FOLFIRI, a safety run-in phase was performed before the randomised phase II, but I wonder how the sample size analysis and power of the study was calculated. Whether this number of sample size is enough to draw the conclusion of the study.
  2. I wonder control groups were selected and whether they were given any palecebo..
  3. I wonder whether authors considered Anthropometric parameters and describe any measures to avoid when there is heterogeneity in the patient samples.

Author Response

REVIEWER #1

The manuscript entitled “Safety of FOLFIRI + Durvalumab +/- Tremelimumab in second line of patients with advanced gastric cancer: a safety run-in from the randomized phase II study DURIGAST PRODIGE 59” by Camille Evrard et al, describes data from the safety run-in phase with FOLFIRI Durvalumab (arm A) or FOLFIRI Durvalumab and Tremelimumab (arm B). Among the 11 patients included, 63.6% experienced at least one grade 3-4 adverse events (AEs) related to the treatment, most 32 frequently neutropenia (36.4%). The authors concluded that this safety run-in phase demonstrates an expected safety profile of FOLFIRI with Durvalumab +/- Tremelimumab combination allowing the randomised phase II.

Overall, this is an interesting concept, however there are some important questions that needs to be addressed to support this study.

We thank the reviewer for highlighting the importance of our manuscript and the relevant comments.

  1. In the present study the DURIGAST PRODIGE 59 study (NCT 03959293) was a randomised, open-label, multicenter, non-comparative, phase II study conducted in France. Patients with advanced gastric or GEJ adenocarcinoma, pre-treated with a fluoropyrimidine plus platinum salt +/- taxane (F+P±T), were randomized 1:1 between FOLFIRI plus Durvalumab (arm A) or FOLFIRI plus Durvalumab plus Tremelimumab (arm B). Due to a lack of data concerning the combination of ICIs plus FOLFIRI, a safety run-in phase was performed before the randomised phase II, but I wonder how the sample size analysis and power of the study was calculated. Whether this number of sample size is enough to draw the conclusion of the study.

There is no statistical hypothesis for this safety run-in phase, which was requested by the French authorities (ANSM, “Agence nationale de sécurité du medicament). The patient’s number was defined with the French authorities. In addition, we have decided to evaluate first combination of FOLFIRI + Durvalumab and then the combination of FOLFIRI + Durvalumab + Tremelimumab. We are agreeing that a total of 11 patients in this safety run-in phase is low to draw definitive conclusion of the safety of this combination. Nevertheless, since there are some safety data with the combination of FOLFOX + Durvalumab + Tremelimumab we think that the combination of FOLFIRI + Durvalumab + Tremelimumab will be safe and we mostly want to demonstrated the absence of severe diarrhea with this combination, which is not the case among the 11 patients (no grade 3-4 diarrhea). This point is now explained in the Discussion section (line 225 – 235).

  1. I wonder control groups were selected and whether they were given any placebo.

There were no specific selection criteria in this safety run-in, inclusion and non-inclusion criteria were the same as the randomized phase II. For the first step of the safety run-in phase 5 patients treated were with FOLFIRI and Durvalumab with no randomization. For the second step of safety run-in phase (FOLFIRI and Durvalumab versus FOLFIRI, Durvalumab and Tremelimumab) 3 patients were randomized per arm. There was no placebo group in this open-labelled study. These points are now explained in the Methods section (line 129-131).

  1. I wonder whether authors considered Anthropometric parameters and describe any measures to avoid when there is heterogeneity in the patient samples.

In order to prevent inclusion of unfit patients with poor anthropometric parameters, patients with Body weight < 30kg and Eastern Cooperative Oncology Group (ECOG) performance status 2 or more were excluded. In addition, in the randomized phase II patient characteristics will be compared to avoid heterogeneity between two groups and the randomization will be done using minimization technique according to the ratio 1:1 and the duration of disease control with previous first-line chemotherapy will be used for the stratification.

Reviewer 2 Report

The study is well conceived and the manuscript is easy to read.

Herein the authors evaluate the safety of ICIs associated with FOLFIRI regime in 2nd-line treatment of gastric or gastro-oesophageal junction (GEJ) adenocarcinoma. The authors recognized that the most limiting parameter is the low number of patients that will be enlarge in the phase II of the multicenter study.  Surely in the continuation of the study there will be necessary some evaluation of immunological parameters to define the patients' responsivness to ICIs.

The methods are explained in details. the patients well characterized.

My only concern is about the real fitting of this study with the main goal of the journal. 

Author Response

REVIEWER #2

  1. The study is well conceived and the manuscript is easy to read.

Herein the authors evaluate the safety of ICIs associated with FOLFIRI regime in 2nd-line treatment of gastric or gastro-oesophageal junction (GEJ) adenocarcinoma. The authors recognized that the most limiting parameter is the low number of patients that will be enlarge in the phase II of the multicenter study.  Surely in the continuation of the study there will be necessary some evaluation of immunological parameters to define the patients' responsiveness to ICIs.

We thank the reviewer for highlighting the importance of our manuscript. As also highlighted by the reviewer 1, the patient’s number of 11 patients in this safety run-in phase is low to draw definitive conclusion of the safety of this combination. With this safety run-in phase requested by the French authorities we mostly want to demonstrated the absence of severe diarrhea with this combination, which is not the case (no grade 3-4 diarrhea). This point is now better explained in the Discussion section (line 225-235). We are totally agreed that it will be mandatory to evaluate immunological parameters to define the patients' responsiveness to ICIs. Indeed, ancillary studies are planned to identify biomarkers of response to immunotherapy, including PD-L1 expression, microsatellite instability, immune scores and microbiota. This point is now better explained in the Discussion section (line 284-288).

  1. The methods are explained in details. the patients well characterized.

My only concern is about the real fitting of this study with the main goal of the journal.

We thank the reviewer for highlighting that the methods and results are well described. The results of this safety run-in phase with immunotherapy fit for the journal since Biomedicines scope included “Novel targets in various therapeutic areas: oncology” and “drug development”. Moreover the paper is submitted to the special issue “Oncoimmunity and Immunotherapy in Solid Tumors” of Biomedicines.
